# The Design and Preparation of Antibacterial Polymer Brushes with Phthalocyanine Pigments

Yu Zhou [1], Kaimin Chen [1,*] , Li Liu [1], Shaoguo Wen [1] and Taijiang Gui [2,*]

[1] College of Chemistry and Chemical Engineering, Shanghai University of Engineering Science, Shanghai 201620, China; zhouyu1579164463@163.com (Y.Z.); lilyliu_dabu@163.com (L.L.); sgwen1@sues.edu.cn (S.W.)

[2] A State Key Laboratory of Marine Coatings, Marine Chemical Research Institute Co., Ltd., Qingdao 266071, China

[*] Correspondence: kmchen@sues.edu.cn (K.C.); tjgui@163.com (T.G.)

**Abstract:** Phthalocyanine pigments have many problems in waterborne coating applications because of their low polarity, poor dispersion in water, and easy agglomeration properties. In order to solve these problems, the phthalocyanine pigments were encapsulated with a copolymer of methyl methacrylate (MMA) and butyl acrylate (BA) by a mini-emulsion polymerization method. The pigments are effectively dispersed in water and have good compatibility with the resin. Concerning the bacterial reproduction and growth problem for the waterborne system, the resin-encapsulated phthalocyanine pigments were further grafted with antibacterial polymer poly(N-(2-hydroxyethyl) acrylamide) (PHEAA) on its surface using the photoemulsion polymerization technique. Comprehensive properties, including centrifugal stability and chromaticity change, were studied. The polymer encapsulation improved the centrifugal stability of the pigment. The thermogravimetric results showed that the residual mass of C.I. Pigment Green 7 (52.30%) was higher than that of C.I. Pigment Blue 15:3 (30.06%), and the sublimation fastness of PG7 was higher. The TEM results revealed that the shape of the PG7 after encapsulation and grafting was more regular than that of PB15:3. The L* of the pigment decreased after encapsulation but then increased after further grafting. The phthalocyanine pigment composite latex had good antibacterial properties after the grafting of PHEAA.

**Keywords:** phthalocyanine pigment; encapsulation; polymer brush; antibacterial

## 1. Introduction

Organic pigments have excellent color expressiveness, photosensitivity, tinting power, and vivid color properties. Thus, they are widely used in fabric printing and dyeing, inkjet printing, paint preparation, and many other applications [1–3]. On the other hand, the drawbacks of organic pigments, such as their poor dispersion, weak weathering resistance, poor stability, uneven particle size, and easy agglomeration properties, limit the application of organic pigments in many fields such as waterborne coatings [4,5]. The commonly used blue organic pigments are mainly phthalocyanine pigments. Phthalocyanine pigments are aromatic macrocyclic organic compounds with eighteen-π electronically stable conjugated aromatic systems. Phthalocyanine itself does not have color, but it can be dissolved in suitable solvents at high temperatures and complexed with heavy metal salts to form macromolecular complexes with bright colors.

C.I. Pigment Blue 15:3 (PB15:3) is a common metallic phthalocyanine compound with a wide range of applications [6]. The chemical structure of PB15:3 endows itself with excellent chemical stability and better weather stability, and it is considered as the best blue pigment among the various organic pigments [7]. Due to the low polarity of PB15:3, it has good lipophilicity, but the hydrophilicity is relatively poor. Thus, it is difficult to be evenly dispersed in the aqueous medium, causing many problems in the application of waterborne systems. On the other hand, C.I. Pigment Green 7 (PG7) has a similar structure

to PB15:3, which can be prepared by replacing the hydrogen atom on the benzene ring of PB15:3 with a chloride atom through a halogenation reaction. Similarly, PG7 faces the same problem in the process of practical application. Therefore, the phthalocyanine pigments should be modified to meet the requirements in aqueous application systems. The chemical structure of PB15:3 and PG7 is shown in Scheme 1.

**Scheme 1.** The chemical structure of (**a**) C.I. Pigment Blue 15:3 and (**b**) C.I. Pigment Green 7.

With the purpose of solving the problem of low polarity and insolubility of phthalocyanine pigments, the surface of PB15:3 is usually modified by adding dispersants. To obtain a more stable system, Hakeim [8] used three different dispersants, anionic emulsifier sodium dodecyl sulfate (SDS), cationic emulsifier hexadecyl trimethyl ammonium bromide (CTAB), and nonionic emulsifier Egyptol BLM (based on nonyl phenol ethoxlate), to modify the surface of PB15:3, respectively. A comprehensive comparison showed that the particle size modified with SDS was smaller, and the particle size did not change with time, indicating a good long-term stability. Additionally, Hakeim [9] et al. compared the dispersion stability of three pigments with different structures, i.e., C.I. Pigment Green 7, C.I. Pigment Yellow 17, and C.I. Pigment Orange 5. They found that the low polarity brought by the eight aromatic rings in PG7 increased hydrophobicity and led to increased interaction with the dispersants, resulting in higher dispersion stability. He [10] et al. synthesized a carboxylate comb copolymer (St-AAMA-PEG) for pigment dispersion and found that the best properties and the smallest particle size could be obtained when PEG600 was used.

Although the pigment can be effectively dispersed in water by using suitable dispersants, it has poor dispersion stability against mechanical vibrations and is prone to aggregation and settling [11]. Furthermore, the dispersant's dose is high, resulting in adverse effects such as poor water and weather resistance. In order to solve the above problems, the pigment could be encapsulated with resin instead of isolated dispersants. Li [12] et al. prepared a fluorescent pigment latex (FPL) by mini-emulsion polymerization using P(MMA-co-BA) encapsulated with fluorescein. Fluorescent pigment latex (FPL) could effectively improve the hand feel and rubbing fastness of cotton fabrics. Wen [13] et al. prepared three-color ink particles by mini-emulsion polymerization and found that the ink system was stable, with a uniform particle size and a high Zeta potential. The presence of a polystyrene shell could effectively improve the aging resistance of pigments.

For waterborne application systems containing organic pigments, antibacterial agents are generally added to avoid the growth and reproduction of bacteria, which leads to spoilage, demulsification, and decreases in the stability properties and service lives of the products. An antibacterial component is introduced typically by the simple physical mixing method, which is convenient and efficient. Zhao [14] et al. prepared coatings by incorporating a synthetic quaternary ammonium methacrylate compound (QAC-2) into a copolymer of methyl methacrylate (MMA) and ethylene glycol dimethacrylate (EGDMA). The antibacterial properties of the coating against Gram-negative Escherichia coli (*E. coli*)

and Gram-positive Staphylococcus epidermidis (*S. epidermidis*) were good. Shevtsova [15] et al. modified halogenated nanotubes (HNTs) with a poly(oligo(ethylene glycol)ethyl ether methacrylate) (POEGMA) brush, and then silver nanoparticle was loaded. This nanomaterial had a good temperature sensitivity effect and could be used in antibacterial applications, especially in biomedical fields. Liu [16] et al. used Ag as the antibacterial nanoparticle immobilized in spherical polyelectrolyte brushes (SPB) composed of poly(N-vinylcarbazole) (PVK) cores and poly (acrylic acid) PAA chain layers. They found that the prepared coatings had good stability and antibacterial properties, but their inhibition time was relatively short, with only 24 h.

Compared with the physical mixing method, the structural antibacterial method is gaining attention and popularity. The antibacterial component is introduced into the resin surface by chemical reaction or coupling. The sustainability of antibacterial effect is better due to well improved distribution of antibacterial component. Furthermore, fewer VOCs (volatile organic compounds) are released since the antibacterial structure cannot be evaporated after film formation. Among the various structural antibacterial structures, quaternary ammonium compounds (QACs) are mostly applied. Usually, QACs kill bacteria by penetrating their alkyl chains into the microbial membrane and changing the phospholipid bilayer, which causes membrane damage and leads to leakage of intracellular components, thus achieving the effect of sterilization [17–19]. Yan [20] et al. prepared a diblock polymer brush on silicon wafers (Si-g-PEGMA-PDMAEMA), and this sample was subjected to quaternization. The observation of bacterial morphology by SEM showed that the quaternized Si-g-PEGMA-PDMAEMA was able to kill bacteria (*E. coli* and *S. aureus*) effectively. Liu [21] et al. prepared polymer brushes consisting of the poly (hydroxyethyl methacrylate) (PHEMA) outer layer and the anionic inner layer loaded with cationic antibacterial peptide (AMP), which can effectively inhibit bacterial adhesion and kill bacteria by exposure to positively charged amines. Ng [22] et al. prepared a brush library by a high-throughput method using the surface-initiated photoinduced electron transfer-reversible addition-fragmentation chain transfer polymerization (SI-PET-RAFT) method to effectively tune the degree of attachment of Gram-negative Pseudomonas aeruginosa (PA) biofilms. Recently, a new type of antibacterial polymer, poly(N-(2-hydroxyethyl) acrylamide (PHEAA), was selected to repel biomolecules, including proteins and bacteria, based on a physical barrier from a hydrated layer [23]. In our previous work, it was also reported that the brush structure, based on PHEAA, provides high antifouling performance, which is also useful to bacteria resistance [24].

To address the problems of the poor dispersion of phthalocyanine pigments and the poor antibacterial effect of waterborne coatings in a cost-effective way, in this study, both PB15:3 and PG7 were selected. Pigments were first encapsulated by a conventional poly(methyl methacrylate-co-butyl acrylate) (P(MMA-co-BA)) copolymer by mini-emulsion polymerization to enhance their dispersion and compatibility properties to other components when used in practical applications. The pigment/resin composite latex was further modified by an antibacterial polymer PHEAA via a photoemulsion polymerization method to ensure a good antibacterial property on the composite latex. The pigment composite was fully characterized by the methods ofdynamic light scattering (DLS), thermogravimetric analysis (TGA), transmission electron microscopy (TEM), and Fourier-transform infrared (FT-IR). The different preparation processes and properties, including the antibacterial properties for PB15:3 and PG7 were compared.

## 2. Materials and Methods

### 2.1. Materials

C.I. Pigment Blue 15:3 (PB15:3) and C.I. Pigment Green 7 (PG7) were supplied by Shanghai Macklin Biochemical Co., Ltd.(Shanghai, China), and their chemical structures are shown in Scheme 1. Methyl methacrylate (MMA), hexadecane (HD), N-(2-hydroxyethyl) acrylamide (HEAA), butyl acrylate (BA), Triton X-100(TX-100), and sodium dodecyl sulfate (SDS) were purchased from Shanghai Titan Scientific Co., Ltd. (Shanghai, China).

Hexadecyl trimethyl ammonium Bromide (CTAB) was supplied by Shanghai Zhanyun Chemical Co., Ltd. (Shanghai, China). Potassium persulfate (KPS) was provided by Aladdin Industrial Co. (Hangzhou, China). Vacuum distillation was used to remove the inhibitor from MMA and BA. HEAA was purified through alumina column chromagraphic separation. The photoinitiator 2[p-(2-hydroxy-2-methylpropiophenone)]-ethylene glycol-methacrylate (HMEM) was prepared, according to the previous method, from 2-hydroxy-40-hydroxyethoxy-2-methylpropiophenone (HMP) (Irgacure 2959) and methacryloyl chloride (MC) [24,25]. LB (Lysogeny broth) broth and LB agar were purchased from Qingdao Hope Bio-technology Co. (Qingdao, China) KCl, NaCl, $NaH_2PO_4$, and $KH_2PO_4$ were supplied by Shanghai Titan Scientific Co., Ltd. (Shanghai, China). Waterborne polyurethane was supplied by Guangzhou Yu Heng Environmental Protection Material Co., Ltd. (Guangzhou, China)

*2.2. Preparation of Pigment Composite Latexes*

First, photoinitiator HMEM was synthesized by a Schotten–Baummann reaction. Typically, 30.0 g of HMP and 13.6 g of MC were mixed in 200 mL of acetone and 10 mL of pyridine. The resulting product was purified through chromatography on silica gel using chloroform/acetone (5:1 in volume) as an eluent. SDS aqueous solution was prepared by adding 0.2 g SDS to 45 g distilled water. Next, 0.5 g pigment was added to a mixture containing 2.5 g MMA, 2.5 g BA, and 0.2 g HD at room temperature to form the oil phase. The oil phase was stirred evenly followed by sonication by the ultrasonic processor (Scientz-IID) for 10 min at 250 W with 1 s pulse on and 1 s pulse off under ice cooling. Then, the oil mixture was added to the SDS aqueous phase and homogenized by ultrasonication for 20 min at 250 W with 1 s pulse on and 1 s pulse off under ice cooling to receive a mini-emulsion. Subsequently, the mini-emulsion was transferred to a three-necked flask containing 5 mL solution that dissolved 0.1 g KPS. Additionally, the mini-emulsion polymerization was carried out at 70 °C for 7 h under a nitrogen atmosphere to obtain polymer-encapsulated pigment (defined as pigment@P(MMA-co-BA)). At the end of polymerization, 5 g photoinitiator (HMEM) acetone solution (The mass ratio of HMEM and acetone is 1:5) was added dropwise in the dark. A thin layer of photoinitiator was slowly grafted onto the pigment composite in 2.5 h. Finally, the obtained products were dialyzed against water to remove the impurities.

*2.3. Preparation of Antibacterial Pigment Composite Latexes*

In total, 5 g purified pigment@P(MMA-co-BA) and 0.27 g of HEAA were added to a homemade photoreactor, and the mixture was diluted to 1.0 wt% by distilled water. Then, the reaction was conducted under light-proof conditions and nitrogen atmosphere by magnetic stirring and UV irradiation (Hg lamp, wavelength: 200–600 nm, power: 150 W) for 1.5 h. The schematic illustration for the preparation of antibacterial pigment composite was shown in Scheme 2.

*2.4. Characterization*

2.4.1. Dynamic Light Scattering (DLS)

The particle size (hydrodynamic diameter) was measured at 25 °C using a dynamic light scattering (Malvern Nano-S90) instrument at a scattering angle of 90°, and the samples were diluted with water to a certain concentration prior to testing. Each sample was measured three times, and the average value was taken with a small standard deviation (SD) of ±1 nm.

2.4.2. Zeta Potential

The pigment composite latexes were diluted 50 times, and the diluted pigments were tested for Zeta potential using a Malvern Zetasizer Nano ZS at 25 °C. Mean value was obtained from triplicates, with a standard deviation of ±1 mV.

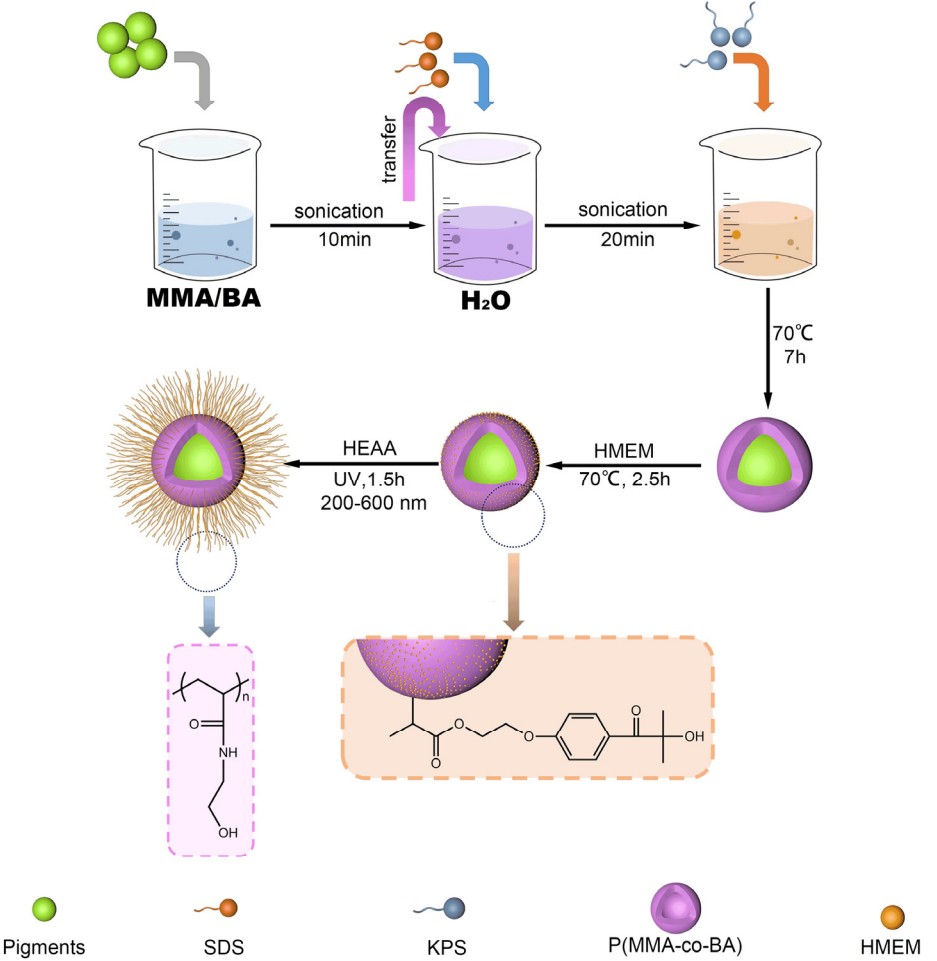

**Scheme 2.** Schematic diagram of the preparation process of pigment@P(MMA-co-BA)@PHEAA.

### 2.4.3. Centrifugal Stability

The maximum absorbance $A_0$ of the sample before centrifugation using a UV spectrophotometer was measured, and the maximum absorption wavelength was recorded as $M_0$. (The maximum absorption peak of PB15:3 was 329 nm and the PG was 320 nm.) Then, the sample was centrifuged at 10,000 rpm for 30 min, 5 μL of the centrifuged sample was diluted 2000 times with 1 mL of distilled water, and the absorbance at $M_0$ was measured and denoted as $A_c$. Centrifugal stability $R$ can be expressed by Equation (1).

$$R = \frac{A_c}{A_0} \times 100\% \tag{1}$$

where $A_0$ is the absorbance before centrifugation, $A_c$ is the absorbance after centrifugation.

### 2.4.4. Fourier-Transform Infrared Spectroscope (FTIR)

FTIR was performed for the dried samples, using the ATR method, by a FTIR spectrometer (AVATAR370, PerkinElmer Instruments Co., Ltd., Shanghai, China).

### 2.4.5. Transmission Electron Microscope (TEM)

The samples were first diluted and then dried naturally at room temperature on a 300-mesh copper grid, and the morphology was observed using a TEM (FEI Tecnai G2 F20) at an operating voltage of 200 kV.

### 2.4.6. Thermogravimetric Analysis (TGA)

The heat loss values of the samples were tested by TGA550 (Newcastle, DE, USA) under nitrogen atmosphere from 30 °C to 800 °C with a heat rate of 10 °C/min.

### 2.4.7. Chromaticity

The samples were diluted to the same ratio, the film was dried with waterborne polyurethane in the ratio of 4:6, and its L*, a*, and b* values were measured using CHN Spec, (CHNSpec Technology (Zhejiang) Co., Ltd., Hangzhou, China).

### 2.5. Antibacterial Experiment

The pigment composites were mixed with aqueous polyurethane in the ratio of 4:6 in mass and then poured into the molds to obtain films after drying at 60 °C. After drying, the samples were placed in a Petri dish and sterilized by UV for 20 min. After the bacterial solution was diluted by PBS buffer, 200 μL of the bacterial solution was added dropwise onto the samples. The samples were then covered with sterilized polyethylene film and transferred to a constant temperature incubator for 24 h at 37 °C. After 24 h, the sample was removed, the surface of the bacterial solution was rinsed with 1.8 mL of PBS buffer to obtain the eluate, and the eluate was vortexed for 30 s. The vortexed eluate (20 μL) was taken, spread evenly on LB medium, and then transferred to the incubator for 24 h at 37 °C.

## 3. Results and Discussion

### 3.1. Preparation of Pigment Composites Latexes

Dispersants are critical to stability and inhibit the agglomeration and flocculation of pigments in the system. Three different dispersants were selected for the dispersion of the pigments, i.e., cationic dispersant, CTAB; non-ionic dispersant, TX-100; and anionic dispersant, SDS. The Zeta potential data of composite latexes prepared by different emulsifiers were shown in Figure 1. The absolute values of the Zeta potential of the PB15:3 and PG7 composite were the highest when using the cationic emulsifier CTAB, which were 25.4 mV and 31.7 mV, respectively. The Zeta potentials of the composites with anionic emulsifier SDSs were −21.2 mV and −24.7 mV, respectively. The higher absolute value of the Zeta potential was mainly due to the fact that the copper in both the PB15:3 and PG7 were coordination atoms, and the coordination copper itself showed a positive charge, which led to electrostatic interaction with the negatively charged SDS, resulting in a lower Zeta potential of the system [3]. The cationic group formed after the dissolution of CTAB, and the coordination copper contents were both positively charged, thus increasing the Zeta potential. In addition, comparing the Zeta potentials of the PB15:3 and PG7 composites, it could be found that the absolute value of Zeta potential was higher for PG7 using all three emulsifiers. The outer ring of the PG7 structure increased its hydrophobicity and improved the interaction with the dispersant [3]. The higher the absolute value of Zeta potential, the more stable the system was and the stronger the electrostatic interaction.

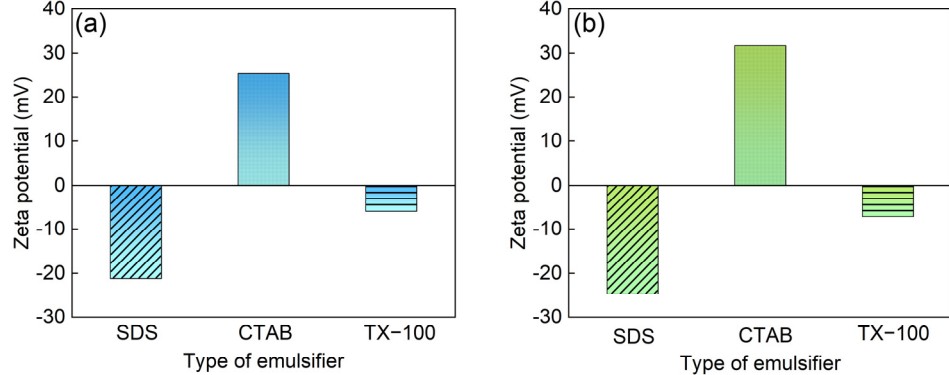

**Figure 1.** Zeta potential of composite latexes prepared by different emulsifiers (**a**) PB15:3, (**b**) PG7.

The dosage of SDS has a crucial effect on particle size. As depicted in Figure 2, with the increase in SDS dosage, the particle size of PB15:3 and PG7 decreased. The particle sizes of PB15:3 and PG7 were the largest, 167 nm and 160 nm, respectively, with 0.1 g of SDS. Additionally, the size was the smallest, 141 nm at 0.2 g of SDS for PB15:3, whereas the smallest was 122 nm at 0.3 g of SDS for PG7. PG7 required more emulsifier than PB15:3 to achieve the smallest particle size, which could have been ascribed to the Cl atoms in the outer ring of the PG7. More emulsifier was needed to provide more hydrophilic groups to combine with the pigment and make the pigment more soluble in water. In order to facilitate the comparison of the PB15:3 and PG7, the particle size was close when the amount of emulsifier was 0.2 g. Thus, the amount of SDS emulsifier was selected as 0.2 g.

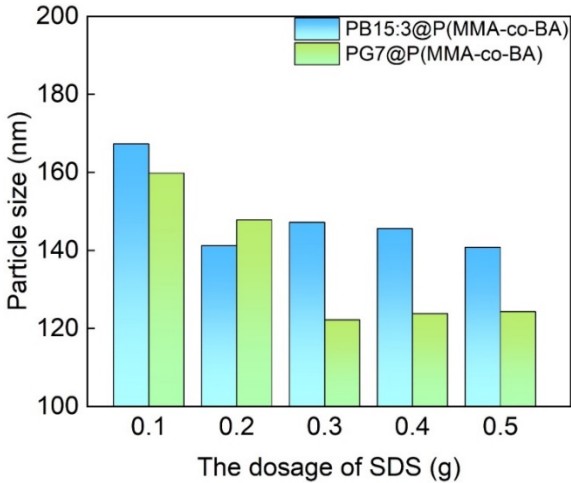

**Figure 2.** Effect of emulsifier dosage on the particle size (diameter) of PB15:3 and PG7 composite latexes.

The centrifugal stability properties of the PB15:3 and PG7 were measured by the specific absorbances before and after centrifugation at 330 nm and 319 nm, respectively. The results are shown in Table 1. The centrifugal stability values were 6.9% and 20.6% for PB15:3 and PG7 and increased to 49.0% and 38.7% after encapsulation with P(MMA-co-BA), respectively. It was clear that the centrifugal stability values of the pigments after encapsulation with polymers was higher than those of the pure pigments. The polymer encapsulation greatly improved the centrifugal stability of the pigment dispersions. This was mainly due to the fact that the polymer on the surface of the pigment increased the spatial resistance and electrostatic repulsion, which made particle deposition difficult. In addition, centrifugal stability was also related to the particle size. Pigments that were not encapsulated with polymers were usually agglomerated and had large particles, which made them very susceptible to settling during high-speed centrifugation. After encapsulating with polymers, the particles became smaller and, therefore, more easily dispersed in water during centrifugation. Regardless of the centrifugal stability of the original pigments, the use of polymer encapsulation could effectively enhance the centrifugal stability of the original pigments.

**Table 1.** Centrifugal stability of original pigments and pigment composite latexes.

| Samples | Diameter (nm) | Centrifugal Stability (*R*) |
| --- | --- | --- |
| PB15:3 | 160 | 6.9% |
| PB15:3 P(MMA-co-BA) | 141 | 49.0% |
| PG7 | 224 | 20.6% |
| PG7@P(MMA-co-BA) | 148 | 38.7% |

### 3.2. Preparation of Functionalized Composite Latexes

FTIR can be used to confirm the successful introduction of added components, and the results are shown in Figure 3. The FTIR spectra of PB15:3, PB15:3@P(MMA-co-BA), and PB15:3@P(MMA-co-BA)@PHEAA are shown in Figure 3a. For PB15:3, The absorption peaks at 1333 cm$^{-1}$ and 1286 cm$^{-1}$ are C-C or C-N stretching vibrations, and the absorption peak at 1165 cm$^{-1}$ is an asymmetric stretching vibration of C-O-C. For PB15:3@P(MMA-co-BA), the new absorption peak at 1731 cm$^{-1}$ is a C=O stretching vibration due to the presence of an ester group, which can confirm the successful encapsulation of PB15:3 particles by P(MMA-co-BA). As for PB15:3@P(MMA-co-BA)@PHEAA, the new absorption peak at 1559 cm$^{-1}$ is a stretching vibrational peak of the N-H amide bond, and the absorption peak at 3426 cm$^{-1}$ is a stretching vibration of N-H and O-H., which confirms the successful grafting of PHEAA. Figure 3b shows the FTIR spectra of PG7 case. For PG7, 1303 cm$^{-1}$ and 1211 cm$^{-1}$ are the stretching vibrations of C-C and C-N. Additionally, the asymmetric stretching vibration of C-O-C appears at 1150 cm$^{-1}$. For PG7@P(MMA-co-BA), a new absorption peak appears at 1732 cm$^{-1}$, which is due to the stretching vibration of the ester group present in the P(MMA-co-BA) copolymer. After the grafting of PHEAA, the peak at 1555 cm$^{-1}$ is the N-H amide bond, and the peak at 3274 cm$^{-1}$ is a stretching vibration of N-H and O-H.

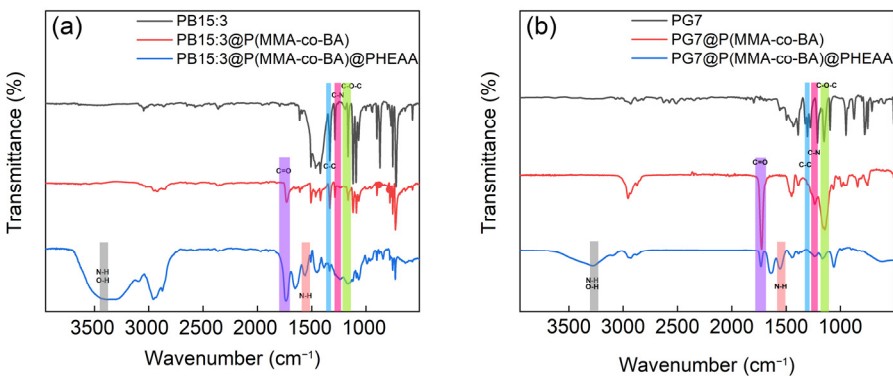

**Figure 3.** FTIR spectra of (**a**) PB15:3, PB15:3@P(MMA-co-BA), and PB15:3@P(MMA-co-BA)@PHEAA; and (**b**) PG7, PG7@P(MMA-co-BA) and PG7@P(MMA-co-BA)@PHEAA.

TEM is a very important and effective analytical method to observe the structures of core–shell particles, which is then used to monitor the size and morphology changes after the encapsulation and functionalization of pigments in this work, as depicted in Figure 4. Figure 4a shows that the shape of the PB15:3 is not uniform, part of which is rod shaped and easy to agglomerate. The size of the particle varies greatly, from 40 nm to 200 nm. The TEM image in Figure 4b demonstrates the PB15:3 encapsulated by P(MMA-co-BA), which shows a distinct spherical shape after encapsulation, and a distinct core–shell structure can be seen. After the grafting of PHEAA, the particle morphology is shown in Figure 4c, which shows that the particle shape is very irregular, and there is a faint shadow around the particle. DLS results show that the particle size increases from 141 nm to 177 nm, which demonstrates the successful grafting of PHEAA onto PB15:3@P(MMA-co-BA).

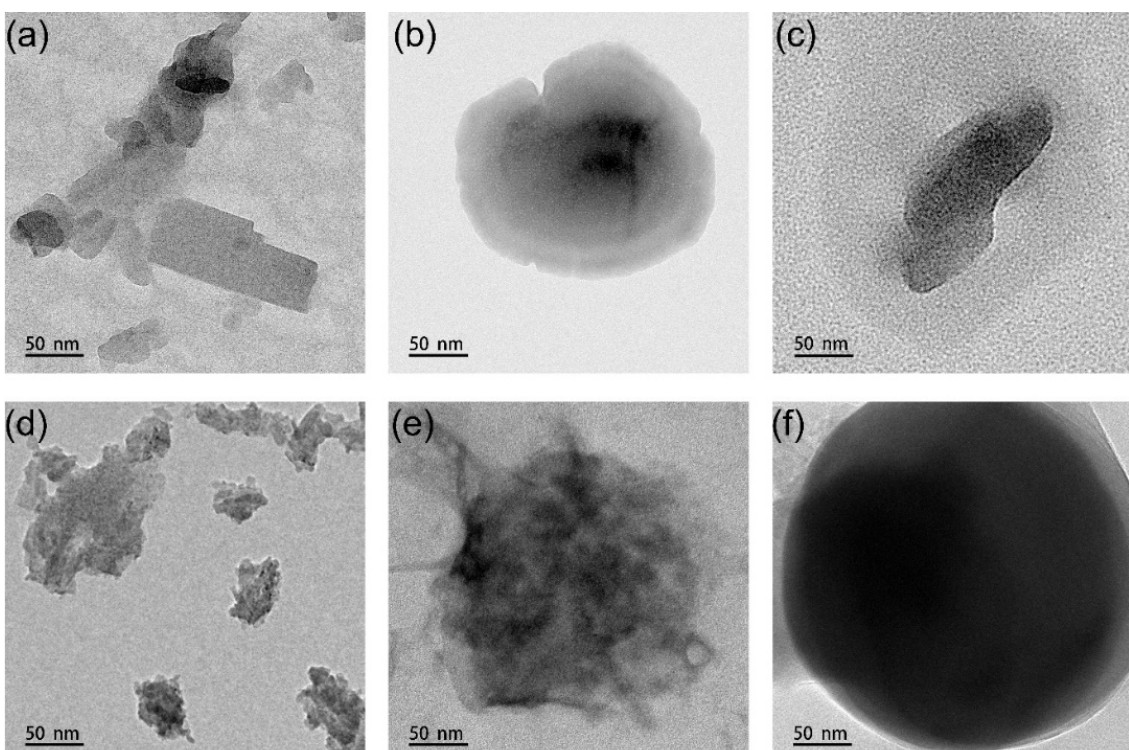

**Figure 4.** TEM images of pigment composites (**a**) PB15:3, (**b**) PB15:3@P(MMA-co-BA), (**c**) PB15:3@P(MMA-co -BA)@PHEAA, (**d**) PG7, (**e**) PG7@P(MMA-co-BA), (**f**) PG7@P(MMA-co-BA)@PHEAA.

Figure 4d shows the morphology of PG7, which has a very irregular shape and is mainly composed of fine particles of PG7 agglomerated in the shape of elliptical rods, with uneven particle sizes. Figure 4e shows the TEM image of PG7 after encapsulating with P(MMA-co-BA). It has an irregular spherical shape, with a particle size of about 150 nm, and no agglomerated PG7 particles can be seen anymore. Figure 4f shows a more regular spherical shape and an evident core–shell structure for PG7@P(MMA-co-BA)@PHEAA. The particle size increases significantly, further confirming the successful grafting of PHEAA onto the surface of PG7@P(MMA-co-BA). The DLS results show that the particle size increased from 148 nm to 185 nm after PHEAA grafting. It is also found that the size is larger for both pigments in TEM compared to DLS results, which is mainly due to the wide distribution of original size of pigments, as shown in Figure 4a,d, and the TEM shown here is not a statistical size.

The shape of PG7 after being encapsulated by P(MMA-co-BA) was more regular than that in the PB15:3 case, which was mainly due to the more uniform size of the unencapsulated PG7. Therefore, PG7 was more likely to form a more regular spherical shape during the encapsulation process, which was consistent with the PDI (polydispersity index) results (The PDI values of PB15:3 and PG7 were 0.302 and 0.231, respectively).

TGA was applied to compare the thermal stability before and after encapsulation and grafting, as shown in Figure 5. In Figure 5a, the mass loss had two stages, with 39.1% loss in the range of 429 °C to 637 °C and 25.1% loss in the range of 637 °C to 763 °C, respectively. The first stage of mass loss was mainly due to the decomposition of aromatic conjugated macrocycles in the structure of PB15:3. The second stage decomposition of PB15:3 was mainly due to the decomposition of the remaining PB15:3 after the completion of the decomposition of the conjugated macrocycles. For PB15:3@P(MMA-co-BA), there was an about 75.5% mass loss in the range of 278 °C and 454 °C, which was mainly due to the decomposition of P(MMA-co-BA) and a small part of the decomposition of phthalocyanine macrocycles. There was an about 13.1% mass loss in the range of 312 °C and 454 °C, which

was the decomposition of the remaining undecomposed phthalocyanine macrocycles. For PB15:3@P(MMA-co-BA)@PHEAA, there was a 75.8% mass loss in the range of 213 °C and 493 °C, and these mass losses were mainly due to the decomposition of P(MMA-co-BA) as well as PHEAA. The 6.4% mass loss between 493 °C and 614 °C was due to the decomposition of the residual PB15:3.

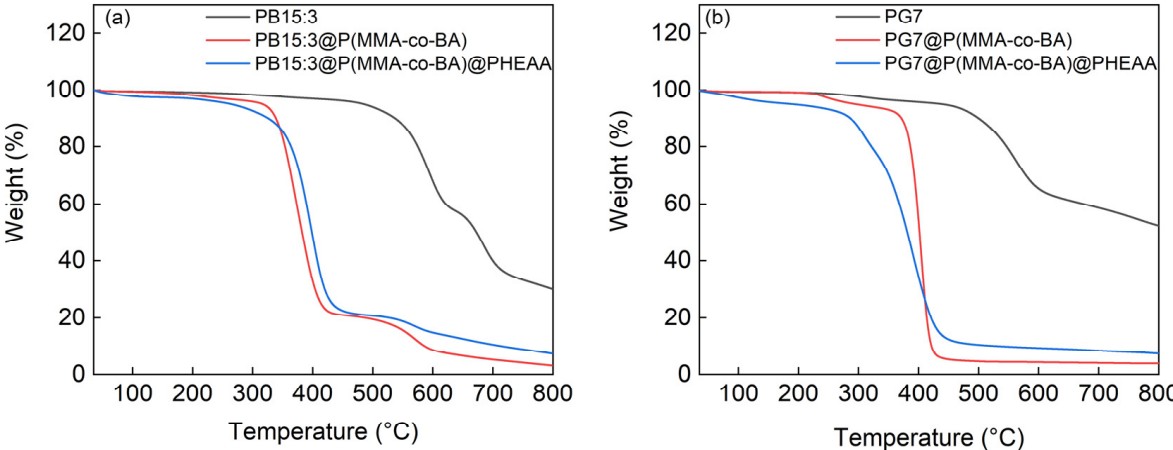

**Figure 5.** TGA of (**a**) PB15:3, PB15:3@P(MMA-co-BA), and PB15:3@P(MMA-co-BA)@PHEAA; and (**b**) PG7, PG7@P(MMA-co-BA), and PG7@P(MMA-co-BA)@PHEAA.

Figure 5b shows the thermogravimetric plots of PG7, PG7@P(MMA-co-BA), and PG7@P(MMA-co-BA)@PHEAA. For PG7, the mass loss was 30.5% at 440 °C to 647 °C, mainly due to the decomposition of the aromatic conjugated macrocycles. For PG7@P(MMA-co-BA), the mass loss was 88.6% at 341 °C to 474 °C. This could have been mainly attributed to the thermal decomposition of P(MMA-co-BA) and contained some degradation of the PG7 during the thermal decomposition process. For PG7@P(MMA-co-BA)@PHEAA, the mass loss was 84.0% between 216 °C and 491 °C, which was mainly due to the thermal decomposition of the PHEAA and P(MMA-co-BA).

From Figure 5a,b, it can be found that the residual masses of PB15:3 and PG7 are 30.1% and 52.3% after heating to 800 °C, respectively. It can be clearly seen that the residual mass of PG7 is higher after thermal decomposition, which is attributable to the presence of Cl atoms at the periphery of the PG7 structure. After encapsulation, the residual mass is 3.10% for the PB15:3 case and 3.85% for the PG7 case. The calculated pigment content is 10% and 8% for PB15:3 and PG7, respectively, which is close to the feed ratio of pigment to resin as 10% in mass. Furthermore, it is evident from Figure 5a,b that the thermal decomposition onset temperatures of PB15:3 and PG7 are lower after encapsulating with P(MMA-co-BA). After the grafting of the PHEAA, the thermal decomposition onset temperature is further reduced. A drop is found before 300 °C in Figure 5b; the mass loss may be ascribed to the oligomer from the previous steps.

### 3.3. Chromaticity of Functionalized Composite Latexes

The color of the pigment has a very important role in the decorative performance of the pigment. The color difference ΔE* was calculated based on L*a*b* data. According to Figure 6, L* decreases after encapsulation with P(MMA-co-BA), and then increases after the grafting of PHEAA. The color of the PB15:3 gradually increases in a* and decreases in b* after encapsulating and grafting, indicating that the color is gradually shifting towards blue green. In contrast, the a* and b* of PG7 increase after encapsulation and slightly decrease after the grafting of the PHEAA, indicating that the P(MMA-co-BA) encapsulation could make the color greener. The grafting of the PHEAA caused a slight decrease in green and a slight increase in the yellow. There is an important relationship between the color difference ΔE* and the visual color difference. A ΔE* value between 1.5 and 3.0 is clearly

perceived. When the ΔE* is in the range of 3.0–6.0, it will be evident. After calculation, the ΔE1* is 2.90 for PB15:3 before and after encapsulation, and ΔE2* is 4.38 after the further grafting of PHEAA. The color difference could also be clearly seen by the optical photos. For PG7, the ΔE1* is 10.66 before and after encapsulation, and the ΔE2* is 5.67 after further grafting. The color difference is more evident for the PG7 case. The main reason for the color difference is that both encapsulation and grafting can change the dispersion of the pigment, which results in a change in chromaticity. Moreover, from the LAB chromaticity space of Figure 6c, it can be seen that the saturation of the pigment blue PB15:3 gradually decreases after encapsulation and grafting. While the saturation of PG7 increases after encapsulation, and a slight decrease occurs after grafting of PHEAA. Chromaticity results indicate that both encapsulation and grafting will affect the L*a*b* data. Furthermore, the trend and range of chromaticity change is closely related to the pigment structure.

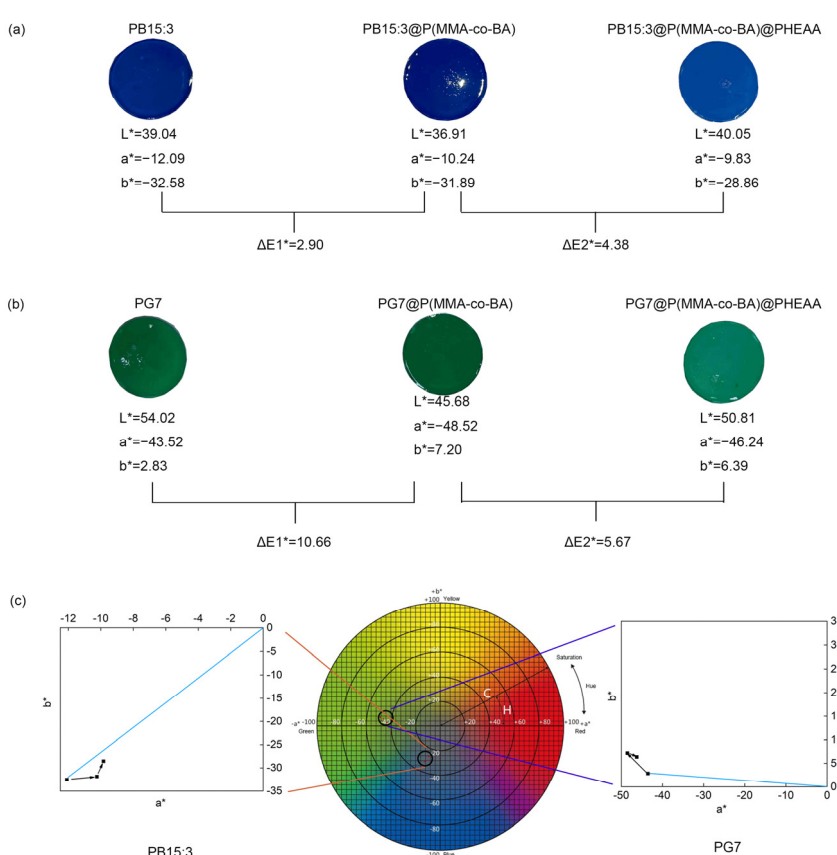

**Figure 6.** The WPU (waterborne polyurethane) film and chromaticity as well as ΔE* of (**a**) PB15:3, PB15:3@P(MMA-co-BA), PB15:3@P(MMA-co-BA)@PHEAA; (**b**) PG7, PG7@P(MMA-co-BA), PG7@P(MMA-co-BA)@PHEAA, (**c**) Hue and chroma variation in PB15:3 and PG7.

### 3.4. Antibacterial Performance of Functionalized Composite Latexes

Gram-negative bacteria Escherichia coli (CMCC(B)4410) (Figure 7a–e) and Gram-positive bacteria Staphylococcus aureus (CMCC(B)26003) (Figure 7f–j) were selected for antibacterial experiments. From Figure 7a,f, it can be seen that *E. coli* and *S. aureus* still have good biological activities and are able to grow on the medium after incubating the eluted bacterial suspensions on the surface of nutrient agar for 24 h. This indicates that WPU films have no antibacterial properties. Similarly, *E. coli* and *S. aureus* are able to grow on PB15:3@P(MMA-co-BA) (Figure 7b,g) and PG7@P(MMA-co-BA) (Figure 7c,h), indicating that PB15:3@P(MMA-co-BA) and PG7@P(MMA-co-BA) do not have any antibacterial properties. When PHEAA is grafted on PB15:3@P(MMA-co-BA) and PG7@P(MMA-co-BA), it can be seen from Figure 7d,e,i,j that neither *S. aureus* nor *E. coli* could grow on the Petri dishes. Functional composite latexes grafted of PHEAA, both for PB15:3 and PG7,

have good antibacterial effects against *S. aureus* and *E. coli*. The antibacterial properties of phthalocyanine pigments are greatly improved after grafting of PHEAA. This may be due to the high resistance of the hydrated layer in the PHEAA structure to bacterial adsorption.

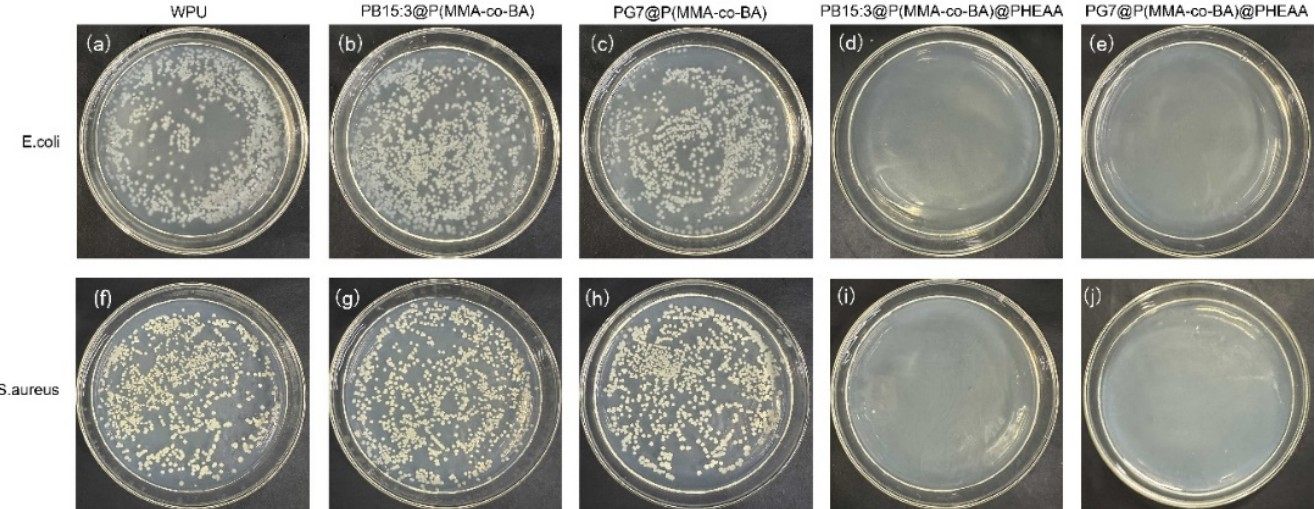

**Figure 7.** The growth of *E. coli* in (**a**) WPU film, (**b**) PB15:3@P(MMA-co-BA), (**c**) PG7@P(MMA-co-BA), (**d**) PB15:3@P(MMA-co-BA)@PHEAA, (**e**) PG7@P(MMA-co-BA)@PHEAA, the growth of *S. aureus* in (**f**) WPU film, (**g**) PB15:3@P(MMA-co-BA), (**h**) PG7@P(MMA-co-BA), (**i**) PB15:3@P(MMA-co-BA)@PHEAA, (**j**) PG7@P(MMA-co-BA)@PHEAA.

## 4. Conclusions

In this work, two kinds of phthalocyanine pigments, C.I. Pigment Blue 15:3 and C.I. Pigment Green 7, were selected and subjected to P(MMA-co-BA) encapsulation and PHEAA grafting. SDS was found to be the most suitable emulsifier for both PB15:3 and PG7 at an optimized dosage, showing good size control and dispersibility properties. PG7 required more emulsifier to reach the minimum particle size compared to PB15:3, due to the presence of hydrophobic Cl atoms surrounding it. The encapsulation of P(MMA-co-BA) could effectively improve the centrifugal stability of phthalocyanine pigments. The TGA results showed that the residual mass of PG7 (52.30%) was higher than that of PG7 (30.06%), and PG7 had a better sublimation fastness compared to the PB15:3 case. The TEM results showed that the shape of PG7 was more regular compared to that of PB15:3 after encapsulation and grafting. After encapsulation, L* decreased, but, after grafting of PHEAA, L* increased. The saturation of PB15:3 decreased after both encapsulation and grafting, whereas the saturation of PG7 decreased after encapsulation with P(MMA-co-BA) and increased slightly after the grafting of PHEAA. The grafting of PHEAA enabled the particles to have good antibacterial properties.

**Author Contributions:** Conceptualization, Y.Z. and K.C.; methodology, Y.Z.; software, Y.Z.; validation, Y.Z., K.C. and L.L.; formal analysis, Y.Z.; investigation, L.L.; resources, S.W.; data curation, Y.Z.; writing—original draft preparation, Y.Z.; writing—review and editing, K.C.; visualization, L.L.; supervision, S.W.; project administration, T.G.; funding acquisition, T.G. All authors have read and agreed to the published version of the manuscript.

**Funding:** This research was funded by State Key Laboratory of Marine Coatings (GZS-002-2020).

**Institutional Review Board Statement:** Not applicable.

**Informed Consent Statement:** Not applicable.

**Data Availability Statement:** Not applicable.

**Conflicts of Interest:** The authors declare no conflict of interest.

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
