# Peer review of "The Design and Preparation of Antibacterial Polymer Brushes with Phthalocyanine Pigments"

_coatings, doi:10.3390/coatings13061114_

Round 1

Reviewer 1 Report

Comments and Suggestions for Authors

Phthalocyanine pigment has long been plagued with challenges in waterborne coating applications due to its low polarity, poor dispersion in water, and propensity for easy agglomeration. However, recent advancements have paved the way for significant improvements in overcoming these limitations. One such breakthrough involves encapsulating the phthalocyanine pigments with a copolymer of methyl methacrylate (MMA) and butyl acrylate (BA) using a mini-emulsion polymerization method. Moreover, in an effort to tackle bacterial reproduction and growth concerns in waterborne systems, the researchers pursued an additional avenue of improvement. They successfully grafted an antibacterial polymer, poly(N-(2-hydroxyethyl) acrylamide) (PHEAA), onto the surface of the resin-encapsulated phthalocyanine pigments using a photoemulsion polymerization technique. In general, the paper presents interesting results and can be accepted for publication after major revision. The following issues should be clarified. 

Information about photoinitiator is almost absent in the paper.  

Why do the authors suggest the fabrication of the poly(N-(2-hydroxyethyl) acrylamide) grafted brushes, it should be explained in detail. Are authors sure that the initiator is integrated into the structure of the pigment composite latexes? please add appropriate information about grafted density.

Please add appropriate information on the mechanism of antibacterial action of the poly(N-(2-hydroxyethyl) acrylamide) grafted brushes.

While the paper highlights improvements in dispersion, compatibility, and antibacterial properties, it lacks a comprehensive analysis of other performance aspects such as adhesion, scratch resistance, and UV stability. Understanding the overall performance of phthalocyanine pigments in waterborne coatings requires a broader evaluation of these factors.

Finally, I suggest citing a highly relevant paper: https://doi.org/10.1016/j.colsurfa.2022.128525

Comments on the Quality of English Language

Minor editing of English language required.

Reviewer 2 Report

Comments and Suggestions for Authors

In this manuscript, authors reported a method to modify pigment composites with polymer coating and antibacterial polymer brushes. The manuscript covers a good amount of works and the writing is well-structured. The work reported here is scientifically sound and of good novelty. This manuscript will be a good contribution to the field. I recommend accepting the manuscript after minor revision. Please see below comments.

Comments:

1. Abstract & Graphical Abstract: Please define the full name of SDS here.

2. Page 2, Line 55: Please provide full names of SDS and CTAB when they first appear in the body text.

3. Page 2, Line 55: What is Egyptol BLM? When I search it online, there is no result related to any chemicals. It would be helpful to readers if authors can add some short explanation, e.g., a trade name of XXX based chemical.

4. Page 2, Line 41-51: I suggest referring Scheme 1 in this paragraph, so that readers can have an immediate understanding of the chemical structures.

5. Page 2, Line 68: “Although the pigment…delamination.” Please add references.

6. Page 2, Line 73: What does “sense of touch” mean? Do authors mean “hand feel”?

7. Page 2, Line 84-85: Species/bacteria names are suggested to be italicized. The same applies to the rest of the manuscript.

8. Page 3, Line 93: Please provide the full name of VOCs.

9. Page 3, Line 90-106: I suggest adding this article as reference where a range of polymer brushes with antibacterial effects were explored. https://doi.org/10.1021/acsami.0c15221

10. Page 3, Line 130: Please provide the full name of LB.

11. Page 3, Line 132: Typo on “water-brone”.

12. Page 4, Line 156: It would be helpful for reproducibility if authors specify which type of UV and what intensity was used. For example, 395 nm, 365 nm, H bulb, or others?

13. Page 9, Line 308: Please provide the full name of PDI.

14. Page 10, Figure 5b: Why is there a drop before 300 °C? Insufficient drying from previous steps?

15. Page 10, Figure 5: I suggest calculating the difference in final weights between coated products and original pigments at 800 °C. This can be good indicators about how much polymers were coated around pigments.

16. Page 10, Line 348: Typo on “P(MMA-BA)”. Missing “-co-” here.

17. Page 11, Line 374: Please provide the full name of WPU.

18. Page 11, Line 391-392: This explanation is not clear enough. In the introduction section, authors mentioned that PHEAA has antibacterial property because it is a quaternary ammonium compound that “kills” bacterial. Here, it sounds like it is attributed to repellence to physical adsorption. Also, is there any reason why authors specify “hydrated layer”. Some further elaboration can be helpful if authors have any hypothesis about the importance of “hydrated layer”.

19. None of the hyperlinks in References work, no matter when I click them or copy and paste them. Please double check.

Comments on the Quality of English Language

Easy to understand. A few typos and confusing words (see comments).

Reviewer 3 Report

Comments and Suggestions for Authors

Zhou et al. reported a method about the surface modification of phthalocyanine pigment. According to a series of characterization method, the results show that the modified pigment has both the good dispersion and antibacterial effect. The characterization methods in this study is comprehensive, and from this perspective the reviewer considers this article to be publishable. However, from the point of view of antibacterial properties, the authors only showed the growth pictures of the bacteria and some quantitative data are missing. The reviewer does not think the results are solid enough. Based on these two considerations, the reviewers concluded that this paper should be reconsider after minor revision.

The author could consider the fluorescence experiments to perform quantitative analysis.

It is still not clear enough that how the relevant parameters influence the antibacterial properties, such as the length of P(MMA-co-BA), i.e. the thickness of the P(MMA-co-BA) layer, and the grafting density of the polymer.

Comments on the Quality of English Language

Page 3 line 116, Transmission electron microscopy (TEM) should be transmission electron microscopy.

Reviewer 4 Report

Comments and Suggestions for Authors

The authors present the preparation and characterization of phthalocyanine pigments encapsulated into polymer brushes with antibacterial activity. The copolymer shell was formed through a miniemulsion copolymerization of methyl methacrylate and butyl acrylate. The surface of the particles was further modified via “grafting from“ technique obtaining brushes from poly(N-(2-hydroxyethyl) acrylamide with antibacterial activity. The pigments-containing particles were characterized by DLS, TEM and thermal analyses. Their antibacterial activity was evaluated on Gram-negative and Gram-positive bacteria. Overall, the manuscript is well-written and might be of interest for the readers of Coatings. However, there are some issues that need to be addressed.

1. The authors should explain all abbreviations at the point of their first use. For example, VOCs (page 3, line 93) and HMEM (page 3, line 129).

2.  Is it possible to control the grafting degree and the grafting density of the brushes when applying the described synthetic methods?

3. The authors claim that they ran the DLS analyses in triplicate. They should provide the standard deviation (SD) values on the corresponding graphs concerning the average particle diameters and the zeta-potentials (Figures 1 and 2). 

4.The term “size” used by the authors should be clarified in the text and in the corresponding figures. It is most likely the particles’ diameter but it could be a radius.

5. Concerning the TEM images it would be more convincing to show more than one particle at lower magnification.

6. The authors should comment in the text what is the reason for such big difference between the results for particle diameters obtained from the DLS measurements and the much larger particles (at least two times larger) shown on the TEM images. 

7. Abstract (line 16): The phrase “…were further grafted of antibacterial polymer…” should be corrected to “…were further grafted with antibacterial polymer…”.

Round 2

Reviewer 1 Report

Comments and Suggestions for Authors

After revision, the quality of the paper was strongly improved only minor revision is needed.

First, please add detailed information on photoinitiator HMEM in the main text or supporting information.

Please add information to scheme 2: time and temperature polymerization of the HMEM; condition of the HEAA polymerization (time and wavelength).

Comments on the Quality of English Language

Minor editing of English language required.
